# An Endogenous Retrovirus Vaccine Encoding an Envelope with a Mutated Immunosuppressive Domain in Combination with Anti-PD1 Treatment Eradicates Established Tumours in Mice

**DOI:** 10.3390/v15040926

**Published:** 2023-04-06

**Authors:** Joana Daradoumis, Emeline Ragonnaud, Isabella Skandorff, Karen Nørgaard Nielsen, Amaia Vergara Bermejo, Anne-Marie Andersson, Silke Schroedel, Christian Thirion, Lasse Neukirch, Peter Johannes Holst

**Affiliations:** 1Department of Immunology and Microbiology, The Panum Institute, University of Copenhagen, Blegdamsvej 3B, 2200 Copenhagen, Denmark; 2InProTher, Bioinnovation Institute, COBIS, Ole Maaløes Vej 3, 2200 Copenhagen, Denmark; 3Department of Biomedical Sciences, The Panum Institute, University of Copenhagen, Blegdamsvej 3B, 2200 Copenhagen, Denmark; 4Sirion Biotech GmbH, 82166 Graefelfing, Germany

**Keywords:** adenoviral vectors, cancer, endogenous retroviruses, murine melanoma-associated retrovirus, virus-like particles, virus-like-vaccines, immunotherapy

## Abstract

Endogenous retroviruses (ERVs) account for 8% of our genome, and, although they are usually silent in healthy tissues, they become reactivated and expressed in pathological conditions such as cancer. Several studies support a functional role of ERVs in tumour development and progression, specifically through their envelope (Env) protein, which contains a region described as an immunosuppressive domain (ISD). We have previously shown that targeting of the murine ERV (MelARV) Env using virus-like vaccine (VLV) technology, consisting of an adenoviral vector encoding virus-like particles (VLPs), induces protection against small tumours in mice. Here, we investigate the potency and efficacy of a novel MelARV VLV with a mutated ISD (ISDmut) that can modify the properties of the adenoviral vaccine-encoded Env protein. We show that the modification of the vaccine’s ISD significantly enhanced T-cell immunogenicity in both prime and prime-boost vaccination regimens. The modified VLV in combination with an α-PD1 checkpoint inhibitor (CPI) exhibited excellent curative efficacy against large established colorectal CT26 tumours in mice. Furthermore, only ISDmut-vaccinated mice that survived CT26 challenge were additionally protected against rechallenge with a triple-negative breast cancer cell line (4T1), showing that our modified VLV provides cross-protection against different tumour types expressing ERV-derived antigens. We envision that translating these findings and technology into human ERVs (HERVs) could provide new treatment opportunities for cancer patients with unmet medical needs.

## 1. Introduction

Cancer is a leading and increasingly prevalent cause of death worldwide and, therefore, the development of new therapies to prevent and cure this disease is a global priority [1]. Despite the encouraging advances in the field of cancer immunotherapy, broadly acting and highly tumour antigen-specific vaccines are still needed, leaving many patients with no other treatment options than palliative care. To offer these patients new therapeutic options, we need to identify new treatment approaches and relevant common tumour antigens to selectively target and eliminate cancer.

Endogenous retroviruses (ERVs) comprise about 8% of the genome in humans [2] and over 10% in mice [3]. These viral elements are relics of ancient retroviral infections that integrated into the genome and were transmitted vertically through the germline, being inherited by successive generations. Although some murine ERVs have remained intact and competent, able to replicate and retrotranspose [4,5], the vast majority of human ERVs (HERVs) have become defective and non-infectious due to the accumulation of nonsense/deleterious mutations. Nevertheless, some HERVs have retained intact open reading frames and thus fully functional and active genes with coding potential. In this way, they have maintained the capacity to produce functional proteins which can, in some cases, assemble into virus-like particles (VLPs) [6,7,8,9]. Despite the biological differences between human and murine ERV reactivation patterns, some, such as the murine leukaemia virus (MuLV) and the human ERV type W, H, R, and FRD, are found within the same gamma retrovirus family type [10], leaving room for possible translation from murine models to humans. Interestingly, ERVs have been found to be involved in both physiological developmental processes and disease [11]. While ERVs are tightly regulated in healthy tissues, they can become reactivated in pathological conditions such as age-associated diseases (e.g., neurological and neuropsychiatric disorders) [12,13], inflammatory and autoimmune diseases [14,15], infectious diseases [16] and cancer [17]. In cancer, aberrantly expressed ERVs have been shown to play a critical role in tumour development, contributing to disease onset and progression [18,19,20].

The fact that ERVs are expressed in a wide range of human [17] and mouse [20,21,22] tumours makes them attractive candidates for developing new cancer immunotherapy approaches. Despite the presence of replicative murine ERVs, mouse models may provide the best available model organism to study the principles of immune targeting of endogenous gammaretroviral antigens on tumours, and in particular to investigate the platform immunogenicity and anti-cancer efficacy of ERV-directed therapies in vivo. Interestingly, several studies have claimed that ERV Env proteins contain a region described as the immunosuppressive domain (ISD) in their transmembrane (TM) subunit, also called p15E, which limits the effectiveness of immune responses to ERV proteins [23,24]. This feature is potentially exploited by ERV-expressing cancers to limit the host immune response, escape immunosurveillance and progress [25,26]. Consequently, the use of technologies that were not powerful enough to overcome Env’s immunosuppressive effect or to enhance Env recognition may explain why several attempts at targeting closely related MuLV sequences have been insufficient to eradicate established tumours in mice [27,28,29,30,31,32,33,34,35,36,37].

To overcome this limitation, we developed a novel vaccination strategy, namely virus-like vaccines (VLVs), which consist of a replication-deficient adenoviral vector encoding VLPs made from the structural proteins of the target antigen. This system has multiple advantages and has been shown to be an efficient vaccination platform inducing potent and robust humoral and cellular responses in the field of HIV and malaria [38,39,40]. For targeting murine ERVs in a murine model system, we chose the murine melanoma-associated retrovirus (MelARV), which is derived from a MuLV provirus that integrated into the murine genome [22]. Recently, we encoded the MelARV group-specific antigen (Gag) and the envelope (Env) in a human adenovirus (Adv) type 5 (Ad5) vector. While the in vivo-expressed Gag protein led to the formation and release of VLPs from transduced cells, the Env protein was surface-expressed and incorporated in secreted VLPs. We showed that vaccination with Ad5-ERV generated decent CD8^+^ T-cell responses (close to 1%) compared to a DNA vaccine, which did not induce ERV-specific T-cell responses or exhibit detectable priming for adenoviral boosters. Notably, such a vaccine was able to eliminate small colorectal tumours in mice and protect cured animals from a rechallenge with a different tumour cell line [41].

In the current study, we attempted to improve the previous published results [41] to achieve better immunogenicity and the enable eradication of large established tumours through the modification of the prototype VLV platform. To do that, we introduced two concurrent mutations (E14R and A20F) in the ISD sequence of the VLV-encoded MelARV Env in order to prevent potential local immunomodulatory effects caused by the vaccine itself. The same modification was previously reported by Schlecht-Louf et al. for the friend murine leukaemia virus to hinder the domain’s putative immunosuppressive activity, while still maintaining the fusion competence of the Env protein [23]. This double mutant vaccine is referred to as Adv-ERV ISDmut in the following, while the prototype vaccine with a wild type (wt) sequence is referred to as Adv-ERV ISDwt.

Here, we show that ISDmut increases the activation of in vitro-transduced mouse bone marrow-derived dendritic cells (BMDCs) and increases vaccine-induced MelARV-specific CD8^+^ T-cell responses up to 3% after a single vaccination and up to 11% in an Adv heterologous prime-boost setting in vivo. This represents a 10-fold increase in immunogenicity compared to the previous DNA-Adv prime-boost regimen published by Neukirch et al. [41]. Furthermore, we demonstrate that therapeutic vaccination with Adv type 19a/64 (Ad19a/64) vector encoding the modified ERV ISDmut in combination with an α-PD1 checkpoint inhibitor (CPI) is capable of eliminating the majority of established colorectal carcinoma CT26 tumours (80%) and extend mouse survival in a challenging curative setting. In addition, we find that only mice that were treated with the mutated ISD vaccine in combination with α-PD1 and that survived CT26 challenge were protected from later challenge with 4T1, a murine triple negative breast cancer cell line, that also expresses a variant of the MuLV Env protein [20].

These findings indicate that mutation of the vaccine’s Env ISD increases an induction of cytotoxic T cells (CTLs), and in conjunction with α-PD1 is capable of eradicating aggressive and established tumours in mice. Furthermore, vaccination with ISDmut offers efficient cross-protection against different cancer types expressing ERV (MelARV or related) antigens. We envision that translating these findings and technology to generate highly immunogenic vaccines against HERVs will have clinical value, providing new treatment opportunities for (recurrent) cancer patients with unmet medical needs.

## 2. Methods

### 2.1. Adenoviral Vaccine Vectors

The expression cassettes encoding the mouse ERV MelARV Gag and Env were synthesized by GenScript Biotech (Rijswijk, The Netherlands). The two viral proteins were linked via a self-cleavable porcine teschovirus-1 2A peptide (P2A). In the ISDmut construct, two point mutations were induced in the 20-amino-acid sequence of the ERV (MelARV) Env ISD using the substitutions E14R and A20F, as described by Schlecht-Louf et al. [23] for the MuLV.

The resulting expression cassettes were cloned into the shuttle vectors pO6-A5-CMV-MCS-SV40-pA and pO6-A19a-CMV-MCS-SV40-pA, respectively. The CMV-GOI-SV40-pA was then transferred via Flp-recombination in *E. coli* into the respective BAC vectors containing the genome of E1/E3-deleted replication-deficient adenovirus serotype 5 (Ad5)- or 19a/64 (Ad19a/64)-based vectors. The system was previously described in Ruzsics et al. [42].

Recombinant viral DNA was then released from the purified BAC-DNA by restriction digest with Pac I. The obtained linear DNA was transfected into modified 293 cells capable of repressing the transgene expression [41] for virus propagation. Viral vectors were released from cells via NaDeoxycholate extraction. Residual free DNA was digested by DNase I. Afterwards, vectors were purified by CsCl gradient ultracentrifugation followed by a buffer exchange to 10 mM Hepes pH 8.0, 2 mM MgCl_2_ 4% Sucrose via PD10 columns (GE). Titration was performed based on the RapidTiter method by means of the detection of infected HEK 293 cells via immunohistochemical staining with anti-hexon antibody (Ad5: Santa Cruz, Ad19a: Novus). Insert integrity was confirmed by PCR amplification of the GOI in DNA purified from the purified vectors. The functionality of the insert was confirmed by the qPCR of infected NIH3T3 or 293 cells.

Vaccines encoding for the modified version of ERV (MelARV) Env ISD were called Ad19a/64-ERV ISDmut and Ad5-ERV ISDmut, while the prototype wt vaccines were named Ad19a/64-ERV ISDwt and Ad5-ERV ISDwt, respectively.

### 2.2. Cell Culture

CT26 (undifferentiated colon carcinoma) and 4T1 (triple negative breast cancer) cell lines, derived from BALB/c mice, were purchased from ATCC (CRL-2638 and CRL-2539, respectively) and were cultured in complete RPMI 1640 GlutaMAX medium (61870044, Thermo Fisher Scientific, Roskilde, Denmark). Vero cells derived from monkey kidney tissue and A549 cells derived from human cancerous lung tissue were obtained from ATCC (CCL-81 and CCL-185). These cell lines were, respectively, cultured in Ham’s F-12 nutrient mix GlutaMAX (31765035, Thermo Fisher Scientific) and DMEM high-glucose GlutaMAX medium (61965059, Thermo Fisher Scientific). All media were supplemented with 10% foetal bovine serum (FBS) (F9665-500ML, Sigma-Aldrich, Søborg, Denmark), 1 mM sodium pyruvate (11360070, Thermo Fisher Scientific) and 1% (*v*/*v*) penicillin-streptomycin (15140122, Thermo Fisher Scientific) and were maintained at 37 °C in 5% CO_2_. All cell lines were authenticated by Eurofins.

### 2.3. Mice

Six- to eight-week-old female BALB/c mice were purchased from Envigo (Scandinavia). Upon arrival, mice were allowed to acclimatize to the animal facility for one week prior to initializing an experiment. All experiments and procedures were performed according to the national guidelines and experimental protocols approved by National Animal Experiments Inspectorate (Dyreforsøgstilsynet) (license number 2016-15-0201-01131). In vivo mouse experiments were performed in BALB/c mice, given that the used tumour models (CT26 and 4T1) and the mapping of the known AH1 murine ERV epitope were established and described in this specific mouse strain [41,43].

### 2.4. Mouse BMDCs 

To obtain mouse dendritic cells (DCs), bone marrow (BM) was isolated from the femur of C57BL/6 mice, followed by lysis of red blood cells. The BM-derived cells were differentiated and matured into DCs in vitro, as described by Jin et al. [44]. In brief, cells were cultured in cell culture dishes (833902500, Hounisen, Skanderborg, Denmark) with RPMI1640 GlutaMAX medium (61870044, Thermo Fisher Scientific) supplemented with 1 mM of sodium pyruvate (11360070, Thermo Fisher Scientific), 10% FBS (F9665-500ML, Sigma-Aldrich), 10 mM of HEPES (15630056, Thermo Fisher Scientific) and 1% MEM non-essential amino acid solution (11140035, Thermo Fisher Scientific). In addition, 20 ng/mL of recombinant murine granulocyte-macrophage colony stimulating factor (GM-CSF) (576302, Biolegend, London, United Kingdom) and 55 mM of 2-mercaptoethanol (31350010, Thermo Scientific) were also added to the medium to expand DC precursors. On day 10, 5 ng/mL of interleukin-4 (IL-4) (574304, Biolegend) was added to promote DC differentiation. On day 13, DC maturation was induced by adding 0.5 µg/mL of cytosine-phosphate-guanine (CpG) (ODN-1668, InvivoGen, Toulouse, France) and 1 µg/mL of lipopolysaccharide (LPS) (L5418, Sigma-Aldrich).

### 2.5. Viral Transduction

The expression of the target proteins upon transduction was characterized in A549 or Vero cells. Two hours after seeding, cells were infected with multiplicity of infection (MOI) of 50 of either Ad5-ERV ISDwt, Ad5-ERV ISDmut, Ad19a/64-ERV ISDwt, Ad19a/64-ERV ISDmut or with an empty/irrelevant Ad5 or Ad19a/64 vector as a negative control. Mouse BMDCs were infected with the same VLVs at 250 MOI and 1000 MOI for Ad5 and Ad19a/64, respectively. Five hours after A549 or Vero cell transduction, the medium was replaced with a new serum-free medium and cells were incubated for 24–48 h at 37 °C and 5% CO_2_.

### 2.6. Transmission Electron Microscopy

Transmission electron microscopy (TEM) was used to visualise vaccine-induced budding and secreted VLPs upon transduction. A549 cells were seeded on Thermanox coverslips (150067, Thermo Fisher) with 1 × 10^6^ cells per well in a 24-well plate and transduced as stated in the viral transduction section. After 24 h, cells were fixed with 2% glutaraldehyde in 0.05 M sodium phosphate buffer (pH 7.2). The samples were rinsed three times in 0.15 M phosphate buffer (pH 7.2) and subsequently postfixed in 1% osmium tetroxide in 0.12 M sodium phosphate buffer (pH 7.2) for 2 h. Samples were dehydrated in graded series of ethanol, transferred to propylene oxide, and embedded in Epon according to standard procedures. Following polymerization, Thermanox coverslips were peeled off. Sections of approximately 60 nm were cut using a Leica UC7 microtome (Leica Microsystems, Wienna, Austria) and collected on copper grids with Formvar supporting the membranes. Following the staining with uranyl acetate and lead citrate, cells were examined with a Philips CM 100 TEM (Philips, Eindhoven, The Netherlands), operated at an accelerating voltage of 80 kV. Digital images were recorded with an OSIS Veleta digital slow scan 2k × 2k CCD camera and the ITEM software package.

### 2.7. Single Cell RNA Sequencing and Analysis

BM-derived mouse DCs were cultured, transduced, and stained with MM2-9B6 and a fixable viability dye, eFlour 780, as described in mouse-bone-marrow-derived DCs, viral transduction, and staining for flow cytometry method sections. Cells from two different mice (M2 and M3) were single-cell-sorted (FACS Melody, BD Biosciences, Allschwil, Switzerland) based on viability (Live^+^) and either the expression of eGFP or positive staining of MM2-9B6 PE into 384-well cell-capture plates containing barcoded primers and mineral oil. Plates were processed at Single Cell Discoveries using the SORT-seq method [45], sequenced and analysed. 

The data from all samples were loaded in R (version 4.2.2) and processed using the Seurat package (version 4.3.0—[46]. Due to the low viability of cells transduced with ISDwt and ISDmut, it was only possible to single-cell-sort a limited number of antigen-positive cells, and furthermore, the RNA quality of these cells was poor. In the eGFP libraries, cells with at least 1000 unique molecular identifiers (UMIs) per cell were included, whereas in the ISDwt and ISDmut libraries, cells down to 150 UMIs per cell were included in the analysis. These thresholds were determined by the UMI count distribution of the empty wells in each plate. The datasets were normalized for sequencing depth per cell and log-transformed using a scaling factor of 10,000 and used for dimensionality reduction and clustering. Cells were clustered into 6 clusters using graph-based clustering, and the original Louvain algorithm was utilized for modularity optimization. Differentially expressed genes between ISDwt and ISDmut populations were calculated using the Wilcoxon rank sum test. To find differences specific to the low-quality clusters, the differential expression analysis was repeated using only ISDwt and ISDmut cells from clusters 1, 2 and 5.

### 2.8. Western Blot

Western blot (WB) was performed as described in Neukirch et al. [41]. Forty-eight hours after viral transduction, Vero cell supernatants were collected and filtered using a 0.2 µM membrane. The supernatants were transferred into open 32 mL thick-walled tubes (355631, Beckman Coulter, Nyon, Switzerland) containing a 20% sucrose cushion. VLPs were concentrated from the supernatant by ultracentrifugation at 82.700× *g* (Beckman Coulter Ti 70 rotor) for 2.5 h. Additionally, cells were harvested and lysed in NP40 lysis buffer (FNN0021, Invitrogen, Waltham, MA, USA) containing protease inhibitor (P8340, Sigma-Aldrich). The protein concentration was determined using the Pierce BCA Protein Assay Kit (23225, Thermo Fisher) following the manufacturer’s instructions. In this process, 5 µg of the protein was mixed with NuPAGE loading buffer (NP0007, Thermo Fisher Scientific) and NuPAGE reducing agent (NP0009, Thermo Fisher Scientific). Samples were run in a NuPAGE 4–12% Bis-Tris gel (NP0323BOX, Thermo Fisher Scientific). After transfer, nitrocellulose membranes (IB23001, Fisher Scientific) were incubated overnight (o/n) with an anti-T2A antibody (to detect MelARV Gag) (crb2005269d, Discovery antibodies) at 1:500 dilution. Bound anti-T2A antibody was visualized by incubating membranes for 1 h with a goat anti-rabbit immunoglobulins (Ig) antibody conjugated to horseradish peroxidase (HRP) (P0448, Dako, Glostrup, Denmark). Detection was performed using SuperSignal West Femto Maximum Sensitivity Substrate (34096, Thermo Scientific).

### 2.9. Staining for Flow Cytometry

Analysis of target protein expression upon cell transduction with the vaccine was performed by means of flow cytometry as described in Neukirch et al. [41]. Vero cells, A549 cells and mouse BMDCs were harvested 24 h post transduction for surface staining. Mouse BMDCs were initially incubated with Fc block (rat anti-mouse CD16/CD32, BD 553141) for 5 min on ice prior to staining. To detect target protein expression on the surface of the infected cells, we used distinct primary monoclonal antibodies against different MelARV Env epitopes at 1:50 dilution in FACS buffer (PBS + 1% bovine serum albumin (BSA) + 0.1% NaN_3_) incubated for 20 min at 4 °C. 19F8 and 4F5 anti-MelARV Env p15E/TM antibodies were kindly provided by George Cianciolo (Duke University Medical Centre). MM2-9B6, MM2-3C6, and MM2-9A3 anti-MelARV Env surface (SU)/gp70 antibodies were kindly supplied by Tsuyoshi Takami (University of Arizona Health Sciences Centre). Target-bound antibodies were labelled with a 1:100-diluted PE- or APC-conjugated secondary goat antibody against mouse IgG (405307 and 405308, Biolegend). In some experiments, eBioscience fixable viability dye eFluor 780 (65-0865-14,Invitrogen) was added at 1:1000 dilution to stain dead cells. Mouse BMDCs were further stained for the surface markers MHC-II (I-A/I-E) (BV510, clone M5/114.15.2 (M5/114), BD biosciences), CD11c (FITC, clone N418, Biolegend), CD86 (BV421, clone GL-1, Biolegend), and CD40 (PerCP-Cy5.5, clone 3/23, Biolegend) at 1:250 dilution in FACS buffer for 20 min at 4 °C. Cells were fixed with 1% paraformaldehyde (PFA). 

MelARV antigen-specific T-cell responses were assessed via intracellular cytokine staining (ICS) as described in Neukirch et al. [41]. Briefly, mouse splenocytes and popliteal lymph nodes (pLN) were mashed and stimulated with 1 μg/mL of the H2-Ld-restricted AH1 peptide, derived from the MelARV Env gp70/SU subunit [21]. Cells were incubated for 5 h in complete RPMI1640 GlutaMAX medium containing 3 µM monensin. Stimulated cells were stained with antibodies for the cell surface markers CD8b (BV421, clone Ly-3, Biolegend) (1:200), CD4 (PE-Cy7, clone RM4-5, Biolegend) (1:800), CD45R/B220 (PerCP-Cy5.5, clone RA3-6B2, Biolegend) (1:200), and CD44 (FITC, clone IM7, Biolegend) (1:100) in FACS buffer for 20 min at 4ºC. Subsequently, activated splenocytes were stained for 20 min at 4 °C with antibodies for intracellular cytokines interferon gamma (IFNγ) (APC, clone XMG1.2, Biolegend), and tumour necrosis factor alpha (TNFα) (PE, clone MP6-XT22, Biolegend) diluted 1:100 in 0.5% saponin (47036-50G-F, Sigma-Aldrich). For regulatory T-cell staining, single cell suspensions from the spleen and pLN were stained with the same cell surface markers as above, with the addition of an anti-CD25 antibody (APC, Clone PC61, Biolegend). Dead cells were discriminated by staining with the fixable viability dye eFluor 780 (65-0865-14, eBioscience) at 1:1000 dilution for 20–30 min at 4 °C. Subsequently, cells were fixed and stained intracellularly with the antibody FoxP3 (PE, clone 259D/C7, BD biosciences) using the transcription factor staining buffer set kit (00-5523-00, eBioscience). Flow data were collected on a Fortessa 3 or 5 instrument (BD Biosciences) or the Attune NxT Cytometer and analysed using FlowJo software (Tree Star, Ashland, OR, USA).

### 2.10. Cytokine Screening of Transduced ERV Mouse BMDCs

Mouse BMDCs were cultured and transduced as described in mouse-bone-marrow-derived DCs and viral transduction method sections. Cytokine levels in cell culture supernatants 24 h after transduction were measured using the V-PLEX Proinflammatory Panel 1 Mouse Kit (V-PLEX K15048D, MSD) according to the manufacturer’s instructions. Supernatant dilutions of 1:5 and 1:50 were analysed for the cytokines IL-1β, IL-4, IL-6, IL-12p70, KC/GRO and TNFα.

### 2.11. Tumour Challenge 

For tumour challenge, cells at about 80% confluency were detached from culture flasks using Versene (15040066, Thermo Fisher Scientific). A total of 5 × 10^5^ CT26 cells were injected subcutaneously (s.c.) in the right flank of BALB/c mice (N = 30), while 1 × 10^4^ 4T1 cells were injected into the left mammary gland of the mice (N = 23). Mice were evenly distributed in treatment groups based on tumour size on the day before treatment initiation. Mice bearing tumours measuring more than 200 mm^3^ on day 9 were excluded. Tumours were measured 3 times per week for length (L) and width (W) and the tumour volume was determined using the formula W × L^2^ × 0.5236 [47]. Mice were culled when tumours reached a size of more than 1000 mm^3^. 

### 2.12. Immunization and Treatment

To assess the vaccines’ immunogenicity in vivo, BALB/c mice (N = 4–5/group) were first primed with either Ad19a/64-ERV ISDwt or ISDmut vaccines. After prime vaccination, some mice (N = 4–5/group) received a boost vaccination with either Ad5-ERV ISDwt or ISDmut vaccines at different time-points. Vaccines were administered s.c. in the foot pad of the right leg (prime vaccination) or the left leg (boost vaccination) with 2 × 10^8^ IFU in 30 μL PBS. For evaluation of the cellular immune response, mice were boosted 7, 14 or 28 days post first injection. For the assessment of the humoral immune response, BALB/c mice (N = 9–10/group) were boosted 30 days after prime. Blood samples were taken on days -1, 30 and 40 and mouse serum containing antibodies was extracted by means of centrifugation (at 800× *g* at 8 °C for 8 min) after coagulation for 2 h at RT or o/n at 4 °C. All mice were sacrificed 10 days after their corresponding boost date. 

In therapeutic studies, a single dose of Ad19a/64-ERV ISDwt, Ad19a/64-ERV ISDmut, Ad5-ERV ISDwt or Ad5-ERV ISDmut was administered to BALB/c mice (N = 9–10/group) 10 days after CT26 tumour challenge. An Ad19a/64 vector encoding an irrelevant transgene was used as a negative control. Vaccines were administered s.c. in the left foot pad contralateral to the tumour challenge (right flank) with 2 × 10^8^ IFU in 30 μL PBS. Additionally, all mice received four intraperitoneal (i.p.) injections of 200 μg of α-PD1 antibody (mpd1-mab15-10, InvivoGen) or anti-β-galactosidase antibody isotype control (bgal-mab15, InvivoGen) in 100 μL of PBS on days 10, 14, 17 and 21 after tumour challenge.

### 2.13. Enzyme Linked Immunosorbent Assay (ELISA)

Vaccine-induced ERV (MelARV)-specific antibodies in the serum of vaccinated mice were measured by ELISA. Nunc maxisorp flat-bottom plates (44-2404-21, Thermo Fisher) were coated o/n at 4 °C with MelARV gp70/SU or p15E/TM Env proteins (manufactured by the antigen discovery team, Centre for Medical Parasitology, University of Copenhagen) at 2 μg/mL in PBS. The following day, plates were covered with blocking buffer containing 0.05% BSA buffer, 2.07% NaCl and 0.05% Tween-20 in PBS for 2 h at RT to avoid nonspecific protein-binding. Subsequently, each mouse serum sample was initially diluted to 1:20 in blocking buffer and incubated on the plate for 1 h at RT in a 2-fold serial dilution. A secondary HRP-conjugated polyclonal goat anti-mouse IgG antibody (P0260, Dako) was added at 1:2000 in blocking buffer and incubated for 1 h at RT. Plates were washed 3 times between each step with PBS containing 2.07% NaCl and 0.1% Tween-20. Bound antibodies were detected using TMB PLUS2 (4395A, Kem-En-Tec Diagnostics, Taastrup, Denmark) solution and the colorimetric reaction was stopped after approximately 8 min by the addition of 0.2 M H_2_SO_4_. Finally, the absorbance at 450 nm was measured using an ELISA plate reader (VersaMax Molecular Devices).

### 2.14. Statistical Analyses

Statistical analyses were performed with GraphPad Prism software (v9). For immunological comparison between two conditions, generally between ISDwt and ISDmut groups, we used the nonparametric, two-tailed Mann–Whitney test. For visual purposes, we used combined bar and dot plot graphs. Dot graphs were used to show both each individual datapoint and box plots to facilitate visual comparison between groups. The log-rank (Mantel–Cox) test was used to compare survival between challenged groups. Statistical significances are indicated by asterisks: * (*p* ≤ 0.05); ** (*p* ≤ 0.01); *** (*p* ≤ 0.001). 

## 3. Results

### 3.1. Characterization of the Virus-like Vaccines

The VLV platform consists of a replication-defective adenoviral vector encoding for the MelARV Gag and Env proteins. To ensure a 1:1 stoichiometric expression, the two viral proteins were linked by a self-cleavable peptide P2A. The Gag protein induces the formation of VLPs when expressed in mammalian cells, while the Env protein is incorporated on their surface as the main target [40]. The p15E/TM subunit of the MelARV Env protein contains a 20-amino-acid sequence with immunoinhibitory properties named the ISD. To prevent the vaccine-encoded Env from being immunosuppressive in the vaccine-transduced cells, we decided to modify the prototype wt vaccine based on Schlecht-Louf’s et al. work on related MuLVs [23]. Therefore, we introduced two point mutations in the ISD sequence to abrogate its immunosuppressive function while maintaining the Env fusogenic function and viral infectivity These modifications consisted of substituting the glutamic acid (E) in position 14 for an arginine (R) and the alanine (A) in position 20 for a phenylalanine (F) (Figure 1A). The prototype wt and the modified mut vaccines were named Ad19a/64 or Ad5-ERV ISDwt (shown in blue) and Ad19a/64 or Ad5-ERV ISDmut (shown in coral), respectively.

To determine if the vaccines can generate and display functional MelARV Env and Gag proteins that assemble into VLPs, we transduced mouse BMDCs, human A549 or African Green Monkey Vero cell lines. The cell lines were chosen due to their ease of transduction with moderate titres of adenoviral vectors, which supports a high level of protein expression with minimal vector-derived cell stress. MelARV Env subunits p15E/TM and gp70/SU were detectable using flow cytometry on the cell surface of A549 cells 24 h post transduction with either Ad19a/64 or Ad5 ERV ISDwt/mut (Figure 1B). The structural protein MelARV Gag was detected by Western blot in cell lysates and VLPs, purified from supernatants of Vero cells, 48h after transduction with the MelARV vaccines (Figure 1C). In addition, budding and secretion of VLPs were visualised by means of transmission electron microscopy (TEM) 48 h after A549 transduction (Figure 1D). These results showed that all of our MelARV vaccines were capable of forming and secreting VLPs and were able to display the target antigen (MelARV Env) on the cell surface upon transduction.

### 3.2. ISDmut Vaccine Increased Mouse BMDC Activation 

To ensure that the previous findings were translatable in the context of a murine model, we first evaluated the effect of the modified VLVs on murine antigen presenting cells. For that, BMDCs from C57BL/6 mice were transduced with either Ad19a/64- or Ad5-ERV ISDwt or ISDmut, or with an empty adenoviral vector as a negative control, and examined after 24 h (Figure 2A). Flow cytometric analysis showed that transduced mouse BMDCs were viable and were able to express the target MelARV Env antigen on their surface. BMDCs transduced with Ad19a/64-ERV ISDmut vaccine (coral) showed decreased MelARV Env expression compared to the prototype (ISDwt) vaccine (blue) (Figure 2B and Appendix A), which may be explained by the differential binding of the chosen antibody against its epitope (Appendix A). Interestingly, after Ad19a/64-ERV ISDmut transduction, live MelARV Env^+^ cells showed an increased expression of CD40 and MHCII compared to cells transduced with the wt ERV, suggesting that impairment of the vaccine ISD increased the activation of mouse BMDCs (Figure 2C and Appendix A). Analysis of the DC supernatant 24 h post infection showed increased levels of interleukin-12 (IL-12p70) when DCs were transduced with Ad5-ERV ISDmut (Figure 2D), suggesting that the modified (ISDmut) vaccine was better at inducing the secretion of pro-inflammatory cytokines, which are known to enhance DC antigen presentation and promote activation of cytotoxic T cells. Furthermore, the MelARV vaccines were able to secrete VLPs from transduced BMDCs, as shown by TEM imaging (Figure 2E). These results indicate that the VLVs were able to transduce several mammalian cell lines, including mouse BMDCs. Additionally, the encoded target proteins were present on cell surfaces and intact VLPs were secreted. Modification of the ISD resulted in increased activation of the mouse BMDCs compared to the wt and in a higher induction of IL-12p70 secretion, potentially leading to augmented T-cell activation and proliferation.

While increased co-stimulation, activation marker expression and IL-12p70 expression are highly consistent with a pro-immune phenotype following the mutation of the ISD, the properties of ISDs have so far only been empirically studied, and cellular mechanisms remain unknown. As our use of adenovirus transduction for the first time revealed an ISD-induced pro-immune phenotypic effect measurable by quantitative protein staining in a cell population, we attempted to study this further. Differences in mouse BMDC activation and cytokine secretion were next investigated after Ad5-ERV ISDwt or ISDmut transduction by cell sorting and bulk RNAseq, which resulted in poor RNA quality of MelARV antigen-sorted cells beyond QC thresholds (not shown). Single-cell RNAseq was next attempted by sorting directly into 384-well plates, but again, the relatively low MelARV^+^ population resulted in low number of live antigen-positive cells, and of those cells that were sorted and sequenced, the RNA quality was again very poor. Unbiased clustering was performed (Appendix A) and investigated for cluster-based quality metrics (Appendix A) and showed MelARV-positive cells enriched in clusters with a high fraction of mitochondrial genes. Unbiased clustering could be ordered into high- and low-quality RNA cells (Appendix A), of which the low-quality cluster contained many MelARV-positive ISDwt cells and almost all MelARV-positive ISDmut cells (Appendix A). Appendix A shows a differential gene expression analysis of genes most upregulated in ISDwt cells compared to ISDmut. These hits were not significant due to the low number of reads and can mostly be considered as dendritic cell housekeeping genes. Overall, the single-cell RNAseq data suggest a rapid turnover and RNA degradation of MelARV antigen-expressing cells, and this is even more accelerated in ISDmut-expressing cells. 

### 3.3. ISDmut Increased Vaccine Induced T Cell Responses

To investigate if the modification of the MelARV ISD would alter vaccine immunogenicity, we designed an in vivo immunogenicity study consisting of a heterologous prime-boost vaccination regimen. BALB/c mice were primed with either the Ad19a/64-ERV ISDwt or the Ad19a/64-ERV ISDmut vaccine. Half of each group were boosted with the respective Ad5 vaccine on day 7, 14 or 28. AH1 (MelARV Env gp70)-specific CD8^+^ T-cell responses were evaluated from the spleen and vaccine-draining pLN 10 days after prime (days 17, 24 and 38, respectively) (Figure 3A). Single immunization with the Ad19a/64-ERV ISDmut significantly increased both the percentage (%) and the absolute number (#) of interferon-γ (IFNγ^+^) producing CD8^+^ T-cells on days 17 and 24 compared to mice vaccinated with Ad19a/64-ERV ISDwt (Figure 3B,C). These responses were further increased when the mice received a second immunization with Ad5-ERV ISDmut. The highest magnitude of CD8^+^ T-cell response was induced when the booster vaccine (Ad5-ERV ISDmut) was given on day 28, which was significantly higher than mice receiving the Ad19a/64- and Ad5-ERV ISDwt prime-boost (Figure 3D). Similar results were observed when looking at the double-positive IFN-γ and tumour necrosis factor-α (TNFα^+^) producing CD8^+^ T cells, in both frequency and number (Appendix A). This showed that the ISDmut significantly increased the functionality of effector MelARV-specific CD8^+^ T-cell responses, which in turn is known to correlate with effective responses, higher immunological tumour control, memory T-cell formation and increased survival [48].

To explore which specific T cells expanded upon vaccination, we plotted the ratio of IFNγ^+^ CD8^+^ CTLs to CD4^+^ T regulatory (T_reg_) cells and showed that ISDmut prime-boost vaccination specifically increased effector CD8^+^ T cells rather than immunosuppressive CD4^+^ T_regs_, for which no measurable expansion was observed (Figure 3E). Analysis of immune responses in the vaccine-draining pLN showed that prime-boost vaccination with ISDmut increased local immune activation as compared to the wt vaccines encoding the intact ISD sequence. While the groups vaccinated with the prototype wt vaccines showed almost no induction of T-cell responses in the pLN, the groups vaccinated with the modified vaccines (ISDmut) presented enhanced AH1 (MelARV)-specific IFNγ^+^ CD8^+^ T-cell responses in all timepoints (Figure 3F). We subsequently normalized the local (pLN) and the systemic (spleen) immune responses and confirmed that the wt sequence is disproportionally poor at inducing local effector T-cell responses in the vaccine draining pLN. This was consistent with the hypothesis that ISD impairment reverted local immunosuppression or increased immune stimulation (Appendix A). 

Additionally, we evaluated the vaccines-induced humoral immune response after an Ad19a/64-ERV ISDwt/mut prime only (day 30) and boost with Ad5-ERV ISDwt/mut (day 40) (Appendix A). Antibody responses against MelARV Env gp70/SU and p15E/TM were modest (Appendix A) and only in some cases did ISDmut show a tendency to be superior at inducing antibodies. Nevertheless, several studies, including this one, support the theory that anti-tumour immunity against murine ERV-expressing cancers is mainly mediated by CD8^+^ CTLs [33,41].

### 3.4. ISDmut Vaccine Synergized with α-PD1 and Eradicated Established Colorectal Tumours in Mice

After showing that the impairment of the vaccine’s ISD enhanced MelARV-specific effector CD8^+^ T cells, we wanted to test if the ISDmut also increased vaccine anti-tumour efficacy in an established murine tumour model. For that purpose, BALB/c mice were challenged s.c. in the flank with a well-characterized murine colorectal carcinoma cell line (CT26). Ten days after tumour challenge, mice received a single immunization of the VLVs in combination with four doses of α-PD1 antibody treatment (Figure 4A and Appendix A). Mice that received the Ad19a/64-ERV ISDmut vaccine together with α-PD1 showed significantly increased survival on day 45 compared to those that received the ISDwt or the control Ad19a/64 vaccine (Figure 4B). Overall, 80% of the mice that were vaccinated with ISDmut showed pronounced tumour regression over time and complete tumour clearance (left coral line graph), in comparison to only 30% and 20% in the ISDwt (middle blue line graph) and control (right grey graph) groups, respectively (Figure 4C). This experiment was repeated with the Ad5-ERV vaccines, revealing the same trend, supporting the suggestion that the mutation of the ISD in our adenoviral vaccines synergized with α-PD1 to increase mouse survival and the eradication of established tumours (Appendix A). 

Overall, these results indicate that α-PD1 by itself and in combination with Ad19a/64-ERV ISDwt had only a modest effect on CT26 tumour control but showed remarkable synergistic effects when combined with the ISD modified vaccine, being able to eliminate most of the established CT26 tumours.

### 3.5. ISDmut Vaccine Induced Cross-Protection against Different Cancer Types

Ultimately, we wanted to investigate whether our VLVs induced long-term on-target immunity against distinct cancer types sharing murine ERV-derived cancer antigens. BALB/c mice that survived the previous CT26 tumour challenge (Figure 4) were rechallenged with 4T1 cells, a triple-negative breast cancer cell line. Tumour growth was measured every 2–3 days over 32 days (primary end point) (Figure 5A). CT26 survivors treated with Ad19a/64-ERV ISDmut and four doses of α-PD1 antibody (coral line) showed a significant increase in survival compared to the other groups (Figure 5B). While CT26 Ad19a/64-ERV ISDmut + α-PD1 survivors showed almost 40% tumour clearance (top right coral line graph), wt and control mice were not able to control tumour growth (bottom graphs) (Figure 5C). 

These results indicate that the Ad19a/64-ERV vaccine with impaired ISD in combination with α-PD1 facilitates growth inhibition of different ERV (MelARV)-expressing cancer types eliminating established tumours in mice. This finding suggests that the combinational treatment could be used in both therapeutic and prophylactic regimens, with the prospect of translation into a human ERV-targeting vaccine.

## 4. Discussion

Several studies have highlighted the association between ERV expression and cancer development and stressed its importance as a biomarker and tumour target [17,20]. A well-known example in humans is the beta retrovirus HERV type K (HERV-K), which has been found in and associated with several cancer types such as breast cancer [49,50], ovarian cancer [51], prostate cancer [52], pancreatic cancer [53], and melanoma [54], among others [55]. Other HERVs from the gamma retroviral family have also been linked to a variety of human malignancies including, but not limited to, breast cancer [56], ovarian cancer (HERV-W) [57], testicular cancer (HERV-W) [58] and colorectal carcinoma (HERV-H) [59]. Therefore, developing an immunotherapy strategy that targets ERVs seems key to specifically eliminating and protecting against a wide range of ERV-expressing cancers. The initial challenge here is to develop an immunogenic platform to generate an immune response powerful enough to overcome ERVs’ immunosuppression or lack of immunogenicity to eliminate ERV-bearing tumours. In this study, we evaluated the immunogenicity and anti-tumour efficacy of our novel modified VLV technology encoding the ERV MelARV viral proteins with point mutations inserted in the ISD of Env. The findings in this study represent the first successful attempt at a vaccination strategy targeting the ubiquitous ERV antigen class that, in conjunction with α-PD1 treatment, is able to eradicate established tumours in mice. 

Several studies have attempted to prevent or eradicate murine ERV-expressing tumours by targeting MelARV or closely related MuLV antigens through different immunotherapeutic strategies. Most of the early studies focused on antibody-mediated effects against tumours using monoclonal antibodies [27,28,29,37]. Others investigated the utilization of adoptive T-cell transfer [30,32], while the most recent efforts focused on the development of diverse vaccination strategies, mainly directed against tumours of the murine colon carcinoma CT26. These vaccination approaches include DNA, DCs and peptides, and modified viruses such as the recombinant vaccinia virus [31,33,34,35,36]. Unfortunately, these strategies seemed insufficient to eradicate established tumours in mice. One major restraint on cancer vaccine efficacy is that endogenous T-cell responses against self-antigens (such as in tumours) tend to be comparably weak, since high affinity T cells are eliminated through tolerogenic mechanisms [34]. Moreover, the expression of the retroviral Env protein containing an ISD with immunomodulatory effects can also affect the host immunity and thus the vaccine efficacy when targeting ERV-expressing cancers [25,26,60]. Therefore, highly immunogenic vaccination platforms, potentially with modifications of the target Env ERV protein, are needed for the successful clearance of established ERV-specific tumours. To overcome the limitations of ERV immunogenicity, we initially designed a novel vaccination approach named VLVs. VLVs consist of adenoviral vectors encoding target antigens that assemble into VLPs. The combination of these two elements has shown great potential as they address both arms of adaptive immunity. While VLPs have been shown to be effective inducers of CD4^+^ helper T cells and antibodies, transduction with viral vectors resembles natural viral infections and elicits CTL responses by presenting antigens in an inflammatory context. Thus, the conjunction in VLVs generates powerful, robust, and specific cellular and humoral immune responses, as demonstrated by Andersson et al. in the context of HIV [38,40]. More recently, Neukirch et al. showed that a VLV targeting murine ERVs could effectively prevent the progression of early growing tumours and eliminate them completely in combination with α-PD1 treatment [41]. Despite these encouraging findings, in Neukirch et al., mice were challenged only 2 days prior to vaccination [41], which does not formally represent a therapeutic treatment of established tumours and thus does not address the vaccine efficacy in a more aggressive and challenging tumour setting, which is one of the most significant challenges in the development of cancer immunotherapies.

Following these results, we explored the strategy of modifying the prototype VLV by introducing two point mutations in the ISD of the vaccine-encoded ERV (MelARV) Env, which was estimated to impair its putative immunosuppressive function based on alignment with other experimentally modified ISD domains [23] (Figure 1A). The study presented here shows that the manipulation of the vaccine-encoded antigen circumvented potential antigen-related tumour-induced immunosuppression or increased immune stimulation to the target antigen and, when combined with α-PD1, effectively eradicated established tumours in mice. These results are highly encouraging as they suggest that the modification of VLVs targeting human endogenous gammaretroviral Env on tumours may provide an effective new type of combination therapy for many patients with advanced malignancies unresponsive to CPI monotherapy. The modification of the Env ISD was based on Schlecht-Louf et al.’s work in the friend murine leukaemia virus, who succeeded in abolishing the ERV Env peptide immunosuppressive function without impairing the fusogenic mechanism of the full length Env [23], a claim that was later challenged by Eksmond et al. [61]. Interestingly, we found lower expression of the MelARV Env upon transduction of A549 cells (Figure 1B) and mouse BMDCs with the ISDmut vaccine compared to the ISDwt (Figure 2B). This observation may be due to the fact that the antibodies of choice bind less efficiently to the mutated ISD, resulting in a decreased MelARV Env detection. Another explanation is that mutations in the ISD could affect the protein structure, leading to increased degradation of the Env protein, and hence to lower surface expression. To clarify this result, we assessed the binding of different available anti-MelARV Env antibodies to their target Env epitopes upon Ad5-ERV ISDwt or ISDmut Vero cell transduction (Appendix A). When targeting specifically the ISD of p15E/TM Env subunit with the 19F8 antibody, we observed a drop in the detection of the targeted epitope on the ISDmut infected cells compared to the ISDwt (Appendix A). This result showed that the mutation modifying the ISD might have induced conformational changes in the targeted epitope and/or the loss of antigen recognition by the antibody and therefore a decrease in its binding capacity. Nevertheless, another anti-p15E/TM antibody (4F5) binding to a different epitope outside of the ISD sequence showed increased Env detection on ISDmut-transduced cells (Appendix A). All antibodies against gp70/SU Env subunit (MM2-9B6, MM2-3C6 and MM2-9A3) showed preferential binding to their target on wt transduced cells (Appendix A). This assay illustrated that the differences observed on antigen expression upon transduction seem to be influenced by the chosen antibody and its specific epitope specificity. Therefore, detection of a given epitope on the cell surface of an infected cell is not necessarily a quantitative indicator of the amount of antigen that is expressed on the surface, but rather a mere confirmation of its expression. 

Mouse BMDCs transduced with the ISDmut vaccine showed increased levels of activation (upregulation of CD40 and MHCII surface markers) within the MelARV-expressing population in comparison to the ISDwt (Figure 2C). Additionally, when screening for cytokines in the supernatant of the transduced mouse BMDCs, we observed higher secretion of the inflammatory cytokine IL-12 by the ISDmut transduced DCs (Figure 2D). This phenotype of increased DC activation with the ISDmut vaccine implied a greater ability for antigen presenting cells to activate and mature upon transduction. This is in accordance with the higher expression of co-stimulatory molecules, pro-inflammatory cytokines, and increased antigen presentation on MHC molecules, which are necessary for T-cell recruitment, priming and expansion [62]. Moreover, the fact that ISDmut influences DC activation makes it unlikely to be an exclusive structural effect (e.g., protein stabilization) but points towards the possibility that mutations in the ISD of Env may also produce an adjuvant effect, enhancing immunogenicity in a cell-intrinsic fashion. Nevertheless, the distinct properties and functions of the gamma retrovirus Env protein have been under debate, as reviewed by Hogan V. et al., and it has been challenging to separate the suggested immunoinhibitory effects of the ISD from its effects on cell fusion, entry and infectivity [63].

Despite being able to transduce DCs and observe increased activation on antigen-expressing DCs, we were unable to identify activated or abrogated pathways by means of single-cell RNAseq (Appendix A). MelARV-expressing cells appeared to die rapidly, particularly in the ISDmut containing groups, resulting in the degradation of RNA. A post-translational cell death pathway seems a likely cause, but it will be difficult to study further without means to isolate the relevant cell population in high numbers before they die. Notably, increased cell death may explain a previous discrepancy in the literature, where Eksmond et al. challenged the infectivity of previously validated ISDmut envelope genes because of the poor virus recovery of ISD-mutated viruses in cell culture assays [61]. This was further supported by in vivo observations, when, after a single immunization with the Ad19a/64-ERV ISDmut, significantly higher IFNγ^+^ CD8^+^ MelARV (AH1) responses (~3%) were elicited compared to the ISDwt (~0.5%) (Figure 3B,C). Similarly, the quality of the CD8^+^ T-cell responses, characterized by the detection of double-positive (IFNγ^+^ TNFα^+^) CD8^+^ T cells (~3% or 4 × 10^5^), was also enhanced upon ISDmut vaccination (Appendix A). Compared to our previously published immunogenicity data, reporting a maximum number of 4 × 10^5^ IFNγ^+^ TNFα^+^ CD8^+^ T-cell responses [41], the current Ad19a/64-ERV vaccine with mutated ISD showed a 4-fold increase in the CD8 polyfunctional response. This suggests that vaccination with Ad19a/64-ERV ISDmut has a greater capacity to activate multifunctional CD8^+^ T-cell responses and potentially memory responses.

Notably, we previously published that DNA/Ad5-ERV ISDwt prime-boost immunization did not increase but rather decreased cellular responses induced by a single immunization with the Ad5 vaccine expressing the wt MelARV sequence [41]. Conversely, we showed here that boosting with Ad5-ERV enhanced all primary Ad19a/64-ERV induced CD8^+^ T-cell responses (Figure 3 and Appendix A). In particular, priming and boosting with the Ad19a/64- and Ad5-modified vaccines containing ISDmut greatly enhanced the magnitude of CD8^+^ T-cell responses, reaching up to 11% of IFNγ^+^ CD8^+^ T-cell responses on day 38 (Figure 3D). This represented a 10-fold increase in immunogenicity compared to the results reported by Neukirch et al. with the prototype vaccine [41]. Thus, the mutated ISD significantly enhanced the potency of the vaccine platform in a single vaccination and heterologous prime-boost regimen when compared to the wt. Moreover, in agreement with the potential immune suppressive nature of the ERV Env, we showed that the poorly immune-stimulating DNA prime was rather detrimental as a prime, at least for our VLVs, while the use of distinct adenoviral vectors encoding the same antigen induced superior immunogenicity against self-tumour antigens. As we are aware of the high prevalence of anti-Ad5 immunity in humans, which limits immunogenicity and translation into the clinic [64,65,66,67], we further investigated alternative vaccine modalities to the Ad5 in combination with the Ad19a/64.

In this study, VLV-induced immune responses against MelARV were predominantly cellular rather than humoral. Neukirch et al. reported low or null levels of CT26 tumour-binding antibodies upon Ad5-ERV ISDwt or DNA vaccination, respectively [41]. Despite this, we also investigated antibody levels after Ad19a/64 prime (only) and prime-boost with Ad19a/64 and Ad5-ERV ISDwt or ISDmut vaccines (Appendix A). Those antibody responses were generally modest against both gp70/SU (Appendix A) and p15E/TM (Appendix A) MelARV Env subunits. A small trend towards higher antibody responses was observed when mice were vaccinated with ISDmut, especially against p15E/TM when only primed and against gp70/SU when boosted (Appendix A). Yet, compared to CD8^+^ T cells, antibodies do not appear to be crucial to efficiently target ERV (MelARV)-expressing cancers [33,41]. Consistently, we showed that the depletion of CD8^+^ T cells in challenged and vaccinated mice abrogated the protective effect of the VLV entirely [41].

For efficacy studies, we chose the commonly used CT26 (wt) tumour model, since it is considered a hot/inflamed tumour with a high potential for responding to CPI, given its permissibility to T-cell infiltration [68]. This cancer model has been used in advanced (Nouscom) [42], or in more incipient (Gritstone) [69] stages of tumour development, as a benchmark for testing neoepitope-directed vaccine technologies prior to clinical trials with encouraging results. We already showed that the prototype Ad5-ERV (ISDwt) vaccine injected two days after CT26 tumour challenge (during early tumour development) with and without α-PD1 treatment resulted in 100% and 70% of tumour free mice, respectively [41]. In this study, we tested a much more aggressive model of CT26 tumours, where BALB/c mice were vaccinated and treated with α-PD1 ten days after the challenge (Figure 4A and Appendix A). While in this setting the Ad5 and Ad19a/64 wt vaccines could not control tumour growth (blue line charts), the vaccines containing ISDmut (coral line charts), in particular Ad19a/64, eradicated 80% of the established CT26 tumours (Figure 4B,C) (Appendix A). D’Alise et al. showed that their neoantigen vaccine (Gad-CT26-31) in combination with anti-PD1 treatment provided complete tumour regression in advanced tumour stages of ~50% of mice and that combinational therapy was required to treat large established subcutaneous CT26 tumours [42]. Here, in the same experimental setting, we were able to show up to 80% of tumour clearance, suggesting that ERVs may be viable alternatives to neoantigens provided that immune responses are induced. Importantly, PD1 blocking antibodies alone (black line charts) reached a 20% tumour control rate (Figure 4C). This was consistent with previously published results where the sole use of PD-1 blocking antibodies partially prevented CT26 tumour growth, yet was not able to ultimately control tumour development in most cases [41,70]. Moreover, the observed results represent the clinical response rate to immune checkpoint blockades as single agents (10–35%) in cancer patients [71], making this mouse model highly relevant for translational efficacy studies. Showing that our VLVs effectively synergized with α-PD1 antibody blockade in a setting that likely mimics the clinical scenario, highlighted the potential transferability of these combinatorial treatments into patients that do not benefit from CPI treatment alone. Treating these patients with our modified VLV might allow the generation of significant levels of cancer specific CTL. In combination with α-PD1 blockade, these can surpass the initial tumour microenvironment inhibition in PDL1 expressing tumours, increasing their treatment response rate and thus improving their clinical outcome.

Considering the broad expression of ERV Env in murine (and human) cancers [17,20], the goal in this study was to show that our vaccination strategy could work both therapeutically (clearing ERV-expressing cancers) and prophylactically (preventing the growth of other cancer types expressing ERVs), by generating potent ERV-specific effector and memory T cell responses. Previously, we demonstrated that mice, which became CT26 tumour-free due to vaccination with Ad5-ERV ISDwt in combination with α-PD1 treatment, were further protected against re-challenge with a 4T1-luc cancer cell line expressing ERV antigens [41]. In the current study, we showed the same principle in a more aggressive high-tumour-burden CT26 model, in which complete protection was only achieved when the VLV carried an impaired ISD in combination with α-PD1 blockade treatment. We observed that only mice treated with the Ad19a/64-ERV ISDmut vaccine and α-PD1 (coral line charts) were fully protected (3/8) after re-challenge with the murine breast cancer cell line 4T1 wt, while none of the other groups, including our prototype (ISDwt) vaccine, showed protection (Figure 5B,C). In comparison to Neukirch et al., the 4T1 cancer cells used in the current study did not contain a luciferase reporter, which could have contributed to the previously observed tumour immune control [41,72]. Although sharing murine ERV antigens, the CT26 colon carcinoma cell line and the 4T1 triple-negative breast cancer cell line should be considered distinct from each other, as they have separate origins and were engrafted in differing tissues. The observation that CT26-controlling immune responses induced in vaccinated mice were likewise able to control 4T1 wt tumours (a more stringent tumour model) highlights the increased efficacy and broad potential of our modified VLVs targeting ERV epitopes that are shared in distinct cancers.

While we have initially proven that our prototype vaccine expressing wt ERV antigens was effective against small growing tumours, mutation of the vaccine’s ISD is needed to overcome the Env-related lack of immunogenicity and effectively clear more aggressive and developed cancers in combinational therapy. ISDmut is also required to provide a stronger immune response that survives the initial tumour elimination and provides cross-protection against different cancer types sharing ERV-derived antigens. This murine proof-of-concept study uniquely shows that a single vaccine design such as Ad19a/64-ERV ISDmut can be used in combination with α-PD1 CPI to effectively eradicate multiple ERV-expressing cancer types and prevent further relapse. Though, the clinical translation of these findings will not be straightforward due to the need to determine the most suitable ERV antigen candidates in humans [17], the overall tumour complexity in humans, and the biological differences between human and murine ERV reactivation patterns [73]. Nevertheless, based on the known expression of HERVs in human cancers and the technical breakthrough of the current study demonstrating increased ERV immunogenicity and potent anti-cancer efficacy, we believe that HERVs are emerging as promising specific targets for immunotherapy of human cancers [17,74].

## Figures and Tables

**Figure 1 viruses-15-00926-f001:**
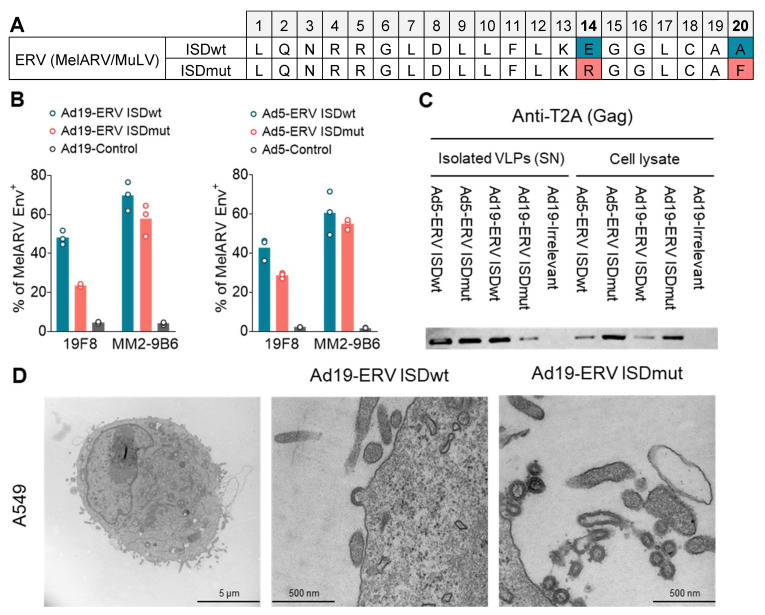
**Vaccine characterization.** (**A**) The prototype vaccine (ISDwt) was modified by substituting E14 R14 and A20 F20 in the ISD of MelARV Env sequence (ISDmut), as described in Schlecht-Louf et al. [23]. (**B**–**D**) Different mammalian cell lines were transduced with Ad19a/64 or Ad5-ERV ISDwt or ISDmut and were analysed for the expression of MelARV Env or Gag, as well as VLP formation and secretion. Adenoviral vaccines encoding an irrelevant transgene or no transgene (empty vector) were used as negative controls when transducing the cell lines. (**B**) Expression of the vaccine-encoded MelARV Env subunits on the surface of A549 cells 24 h post transduction. The primary antibodies 19F8 and MM2-9B6 were used to detect the respective Env subunits p15E/TM (ISD) and gp70/SU, by flow cytometry. (**C**) Expression and release of the vaccine-encoded MelARV Gag protein (detected by anti-P2A) as shown by Western blot after 48 h in cell lysates and VLPs purified form cell culture supernatants of transduced Vero cells. (**D**) Visualization of budding VLPs (circles of approximately 100 nm) in A549 cells at 48 h after transduction with Ad19a/64-ERV vaccines. Images were generated by TEM. Ad19a/64 is referred as Ad19 in the figure.

**Figure 2 viruses-15-00926-f002:**
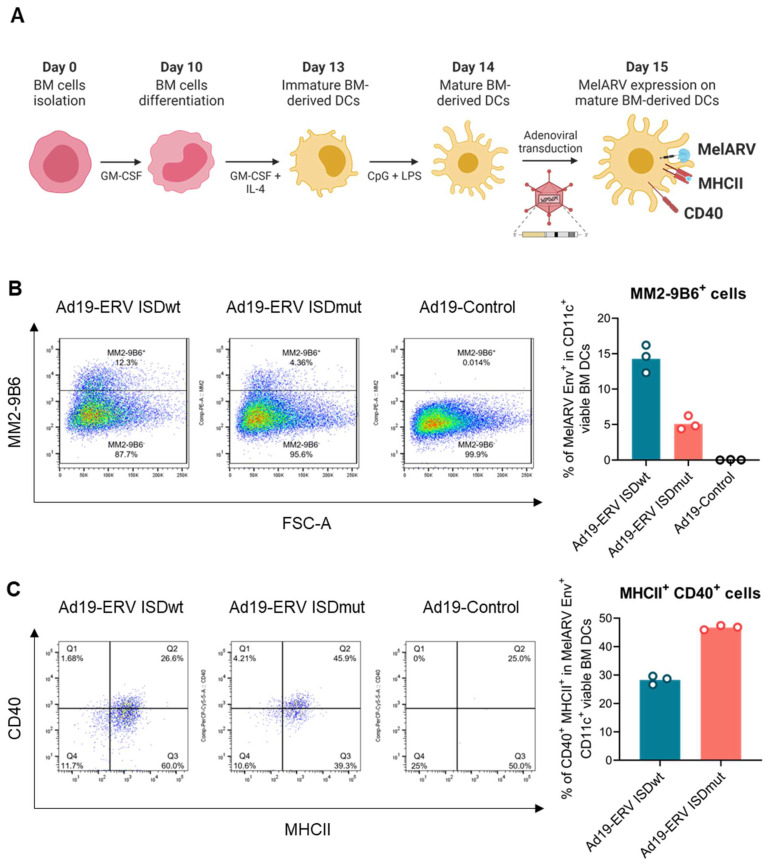
**Impairment of the vaccine ISD increases the activation of BMDCs.** (**A**) In vitro generation of mature DCs from murine BM cells. BM cells were isolated and differentiated into immature DCs by adding GM-CSF and IL-4 into the medium. DCs were matured by adding CpG and LPS in the culture and transduced at 1000 MOI or 250 MOI with the Ad19a/64- or Ad5-ERV ISDwt or ISDmut, respectively. Ad5- or Ad19a/64-empty vectors were used as negative controls. Twenty-four hours post transduction, the expression of MelARV Env and DC maturation/activation markers was assessed by means of flow cytometry. (**B**) Expression of MelARV Env gp70/SU subunit on the surface of transduced live DCs was assessed by means of flow cytometry using MM2-9B6 d. (**C**) Activation/maturation of live MelARV Env^+^ BMDCs was established by measuring MHCII and CD40 co-expression (fraction of double positive cells) using flow cytometry. (**D**) Secretion of the cytokines IL-1β, IL-4, IL-6, KC-GRO, TNF-α, and IL-12p70 was assessed in the supernatant of transduced mouse BMDCs using a V-PLEX assay. Data points represent samples from 6 independent mice. (**E**) VLPs (of approximately 100 nm size), secreted from mouse BMDCs, were visualized by TEM, 24 h after transduction. Ad19a/64 is referred to as Ad19 in the figure. **: *p* < 0.01—Mann–Whitney U test.

**Figure 3 viruses-15-00926-f003:**
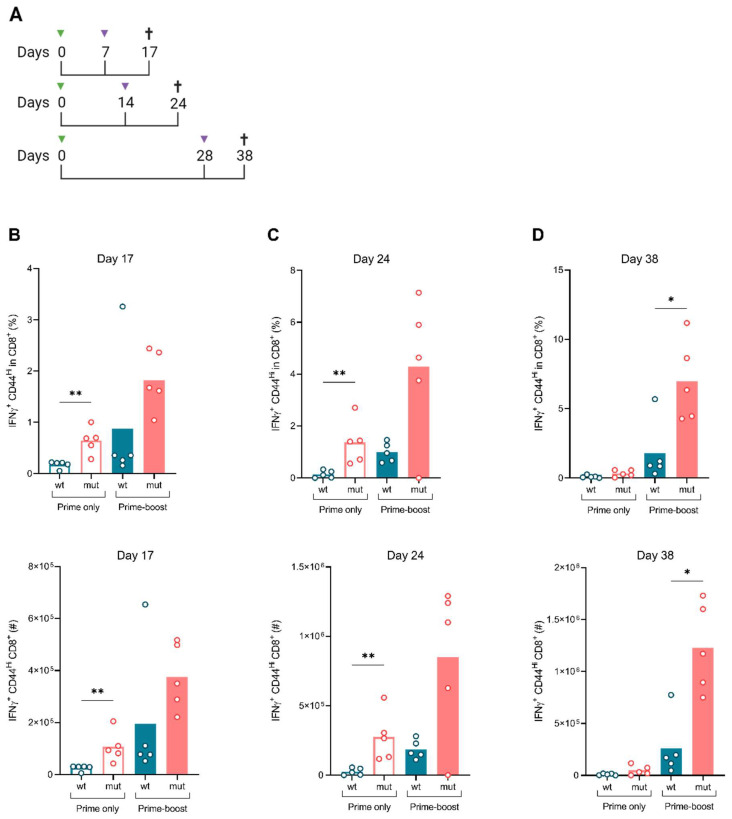
**ISDmut vaccines elicit stronger CD8^+^ T cell responses both locally and systemically.** (**A**) Schematic representation of the vaccine regimen. BALB/c mice were vaccinated s.c. (right paw) on day 0 with either the Ad19a/64-ERV ISDwt or the Ad19a/64-ERV ISDmut vaccine. Half of the primed mice were vaccinated s.c. (left paw) a second time (boost) with either the Ad5-ERV ISDwt or with the Ad5-ERV ISDmut on day 7, 14 or 28. Spleens and vaccine-draining pLN (at the site of booster injection) were collected 10 days after the boost to perform ICS against the MelARV Env gp70/SU H2-Ld-restricted T cell peptide AH1. (**B**–**D**) Immune profiling of spleens in Ad19a/64 +/− Ad5-ERV ISDwt- or ISDmu-vaccinated mice. Frequency (**upper graphs**) and absolute number (**bottom graphs**) of IFNγ^+^ CD44^Hi^ CD8^+^ T cells responding to AH1 MelARV peptide at day 17, 24 and 38 post prime. Day 17 was repeated in two independent experiments. (**E**) Ratio of the percentage of IFNγ^+^ CD44^Hi^ in CD8^+^ T cells and the percentage of FoxP3^+^ CD25^+^ in CD4^+^ T cells (T_regs_) in the spleen of prime-boosted mice. (**F**) Immune profiling of draining pLN in prime-boost (Ad19a/64- and Ad5-ERV ISDwt or ISDmut)-vaccinated mice showing the frequency of IFNγ^+^ CD44^Hi^ CD8^+^ T cells responding to AH1 MelARV peptide at day 17, 24 and 38 after prime. N = 4–5, *: *p* < 0.05, **: *p* < 0.01—Mann–Whitney U test. The gating strategy can be found in Appendix A.

**Figure 4 viruses-15-00926-f004:**
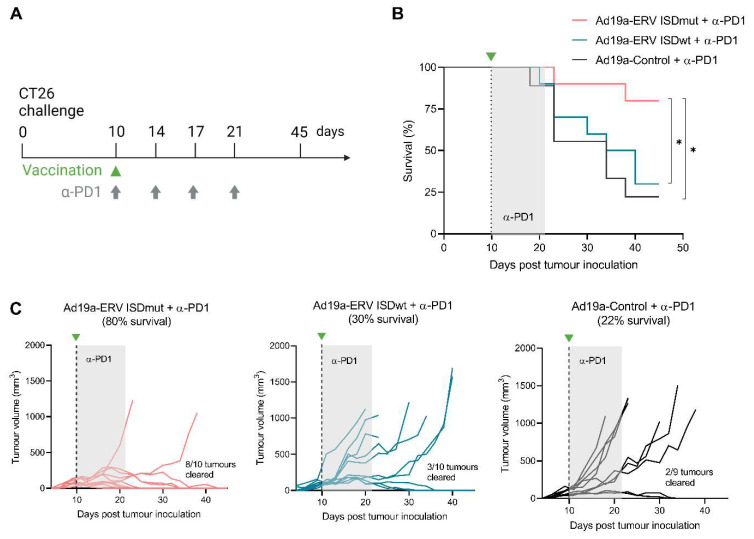
**Ad19a/64-ERV ISDmut synergises with α-PD1 check point inhibitor and eradicates established colorectal tumours in mice**. (**A**) Schematic representation of the tumour challenge and the therapeutic treatment. BALB/c mice were challenged s.c. (right flank) with 5 × 10^5^ CT26 cells and tumour size was evaluated every 2–3 days. Mice were randomized and vaccinated s.c. (left paw) on day 10 with either the Ad19a/64-ERV ISDwt (blue), the Ad19a/64-ERV ISDmut (coral) or the Ad19a/64-Irrelevant (black) vaccine. α-PD1 treatment was administered i.p. concomitant to the vaccination, and then repeated three times every 3–4 days. (**B**) Fraction of surviving mice over time after CT26 challenge and treatment. (**C**) Tumour volume (mm^3^) over time after challenge and treatment. N = 9–10, *: *p* < 0.05—Log-rank (Mantel–Cox) test. Ad19a/64 is referred to as Ad19a in the figure.

**Figure 5 viruses-15-00926-f005:**
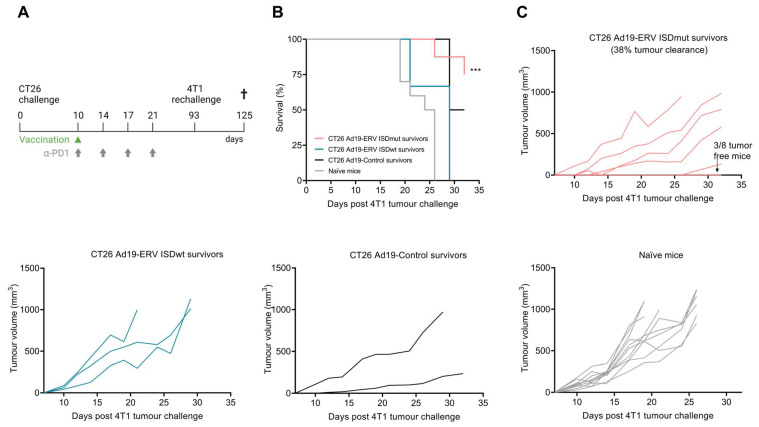
**Ad19a/64-ERV ISDmut in combination with α-PD1 protects against rechallenge with a triple-negative breast cancer cell line**. (**A**) Schematic representation of the tumour challenge and the therapeutic treatment. Ad19a/64-vaccinated BALB/c mice that survived CT26 challenge and naïve mice were (re)challenged with 1 × 10^4^ 4T1 tumour cells into the (left) thoracic mammary fat pad. Tumours were measured every 2–3 days. (**B**) Fraction of surviving mice over time after 4T1 rechallenge. (**C**) Measurements of tumour volume (mm^3^) over time after 4T1 rechallenge. N = 2–10, ***: *p* < 0.001—Log-rank (Mantel–Cox) test. Ad19a/64 is referred as Ad19 in the figure.

## Data Availability

Data is contained within the article or Appendix A. The raw data is available upon reasonable request from the corresponding author.

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
