# Peer review of "An Endogenous Retrovirus Vaccine Encoding an Envelope with a Mutated Immunosuppressive Domain in Combination with Anti-PD1 Treatment Eradicates Established Tumours in Mice"

_viruses, 2023, doi:10.3390/v15040926_

Round 1
Reviewer 1 Report
Despite the numerous successes achieved, cancer therapy still represents a topic of great interest in medical practice. Many studies have identified tumor immunotherapy as a therapeutic approach aimed at replacing chemotherapy.
The study proposed here aims to achieve a very ambitious goal: not only to eradicate a consolidated tumor but also to prevent the development of a second tumor, with the same antigenic characteristics, in a different location.
In this study the authors evaluated the immunogenicity and anti-tumour efficacy of a novel virus-like vaccines encoding the ERV MelARV viral proteins in a mouse model.
The authors have a good experience in the developmental of a vaccination strategy based on a replication-deficient adenoviral vector encoding virus-like particles made of the structural Gag and Env proteins of the target antigen. Notably, they previously demonstrated the efficacy of adenovirus based virus-like vaccines targeting the murine ERV (MelARV) env to prevent growth and progression of colorectal cancer in mice after a single vaccination before or after tumor challenge. The efficacy of the vaccination, increased by the combination with anti-PD-1 treatment, led to complete tumor eradication. Furthermore, the immune response induced by the vaccination was able to protect the animals also from a rechallenge with a different tumor cell line.
Based on the results highlight in a previous paper published by some of the authors (Oncotarget, 2019, Vol. 10, (No. 14), pp: 1458-1472) in the current study the authors attempted to improve their virus-like vaccine platform. In particular, as an innovative element, they inserted two point mutations in the envelop-mediated immunosuppressive domain, in order to prevent the immunosuppressive activity demonstrated by the vaccine itself.
As the authors themselves declare "The same modification was previously reported by Schlecht-Louf et al. “(G. Schlecht-Louf et al., Proc. Natl. Acad. Sci. U. S. A., vol. 107, no. 8, pp. 3782–7, Feb. 2010). Therefore, this modification, while improving the results, cannot be interpreted as an element of originality.
In the previous work the rationale on which the use of HERVs was based as possible candidates for an immunotherapy in tumors was well described. In the present study, however, any references has been omitted, creating, in my opinion, difficulties of interpretation by the reader. Furthermore, the claim that HERV expression is mostly limited to tumor tissues is not entirely correct. Indeed, HERV expression is tightly controlled in normal adult tissues, but is aberrantly expressed not only in cancer but also in inflammatory and autoimmune diseases, aging, type 1 diabetes, neurological and neuropsychiatric disorders, and even in infectious diseases.
Therefore, I think that the introduction should be implemented.
The data on the eradication of colorectal cancer in mice by the synergistic mutant vaccine with anti-PD1 is not entirely new. The same authors write in the abstract of their work published 4 years ago that "A combination with the anti-PD-1 checkpoint inhibitor further increased the efficacy of the vaccination leading to complete tumor regression."
Therefore, I believe that the indication contained in the title of this manuscript "Synergizes with CPI and Eradicates Established Tumors in Mice" should be omitted as it does not characterize the present study.
The demonstration that the mutated vaccine induces cross-protection against different types of cancer is also not a new result.
Unfortunately, this study largely repeats what has already been published, even if it presents numerous points of interest, such as the production of viral particles following the transduction of cells by the vaccine (see TEM), and above all the deepening of the immune response by of the host.
In fact, the study on immunoprofiling is new, interesting and well conducted, aimed at demonstrating the global involvement of the new vaccine in activating the immune system. Being a work on tumor immunotherapy, in my opinion this part should have been highlighted while inexplicably it was included in the supplementary materials
Minor revisions
Fig. 1 panel D. The size bar is not visible in the image
For in vivo experiments were used BALB/c mice while dendritic cells were isolated from the bone marrow of C57BL/6 mice. The rationale should be explained.
The number of animals challenged, immunized and treated is not indicated in materials and methods section.
Should be also explained why the viral transduction was performed on A549 and Vero cell lines.
In conclusion, my suggestion is to set up a new manuscript taking into account the data previously described as a reference, and highlighting only the innovative data.
Author Response
"Please see the attachment."

Reviewer 2 Report
The paper "An Endogenous Retrovirus Vaccine Encoding an Envelope with a Mutated Immunosuppressive Domain Synergizes with CPI and Eradicates Established Tumours in Mice"
Describes the immunisation of mice with constructs containing murine leukaemia virus envelope antigens inducing immune responses and regression of murine tumour cell lines that (presumably) express MuLV antigens.
The basic experimental work seems ok though there are few bits that need to be sorted out , including:
Mann-whitney-U tests (pairwise comparison) are inappropriate for your data as 3-4 conditions in most of your graphs. These should have been analysed by anova (or non-parametric equivalent). The use of the combined bar and dot plot graphs is also a bit misleading and these would have been better presented as a box and whisker type blot showing standard error (as some of your data sets have a very large spread)
your single cell RNAseq data did not produce very meaningful results so should probably be removed
The main problem with the paper is however is that its underlying premise of using Murine leukaemia virus antigens in murine tumours as a model for human endogenous retroviral involvement in human tumours is a bit dubious. The two systems are very different in that mice have active recent retroviruses in their genomes that undergo a full lifecycle with insertional mutagenesis and production of viral particles. The human endogenous retroviruses are not the same classes of virus in the first place and are mostly quite old/mutated/inactive and don't come anywhere near completing a full life cycle. Expression of HERVs in human cells is cell line and disease condition specific (and varies for different families and different loci within the same family). Some decent recent reviews are here:
https://onlinelibrary.wiley.com/doi/10.1002/jmv.28350
https://pubmed.ncbi.nlm.nih.gov/36256614/
some reviews on MuLVs are here:
https://pubmed.ncbi.nlm.nih.gov/18818872/
https://pubmed.ncbi.nlm.nih.gov/22703977/
The references quoted for the effects of the "immunosuppresive domain" are also quite old and cherry picked. There is quite a bit of controversy/doubt over whether this short peptide sequence actually does anything specifically immunosuppressive or whether the initial studies were just demonstrating the effects of a small peptide sticking to larger proteins. A good recent review of the current state of knowledge of this is here:
https://www.mdpi.com/1999-4915/15/2/274
This domain is more useful as a conserved feature of viral fusion proteins than a specific immunosuppresive feature.
As such you've demonstrated a modified construct of MuLVenv sequences produces immune responses to murine cell lines. But its really not clear that the immunosuppressive domain plays any part in this or that the results are remotely translatable to human cell lines/cancers as the HERV complement is so different to mice
Author Response
"Please see the attachment."

Reviewer 3 Report
The work presented is highly relevant for the field, since there is scarce investigation on the effects of the ISD in tumor immune scape. However, the research was conducted once tumors were already implanted and in presence of an additional therapy using the anti-PD1 immune checkpoint. Thus, remains open the prospective protection against tumors after the development of a previous immunity using VLV containing mutated ISD and in absence of any immune checkpoint modulation.
Author Response
"Please see the attachment."

Round 2
Reviewer 1 Report
I acknowledge that the authors made a significant effort to respond to reviewers' requests. With great pleasure I admit that the manuscript in its current state is, in my opinion, greatly improved not only in form but also in its attractiveness for the reader. The introduction has been properly implemented and all suggested clarifications have been properly inserted.
I therefore believe that the manuscript can be accepted for publication in its current form. Just two small suggestions: in the caption of fig.3 write (F) in bold and move the reference to the supplementary figure to the end of the sentence.
Author Response
We are very grateful to see that the effort put into the revision of the manuscript has been successful. We appreciate the feedback received since we believe that it has helped us to improve the disclosure quality of this work making it more accurate and informative for the reader. Finally, we made sure to incorporate the suggestions stated above in the new version of the manuscript.
Reviewer 2 Report
The authors have provided a thorough response and corrections to the paper. The laboratory work appears sound and the results reasonable. I think we will have to agree to disagree on the suitability of mouse ERVs as models for human ones but the discussion has been modified suitably to tone down the conclusions and I am happy enough for this to proceed to publication now.
Author Response
We appreciate the acknowledgment of the effort put into the answers and the modifications in the manuscript. We appreciate the support in proceeding with the publication.